# Hospital Length of Stay and Surgery among European Children with Rare Structural Congenital Anomalies—A Population-Based Data Linkage Study

**DOI:** 10.3390/ijerph20054387

**Published:** 2023-03-01

**Authors:** Ester Garne, Joachim Tan, Mads Damkjaer, Elisa Ballardini, Clara Cavero-Carbonell, Alessio Coi, Laura Garcia-Villodre, Mika Gissler, Joanne Given, Anna Heino, Sue Jordan, Elizabeth Limb, Maria Loane, Amanda J. Neville, Anna Pierini, Anke Rissmann, David Tucker, Stine Kjaer Urhoj, Joan Morris

**Affiliations:** 1Department of Paediatrics and Adolescent Medicine, Lillebaelt Hospital, University Hospital of Southern Denmark, 6000 Kolding, Denmark; 2Population Health Research Institute, St George’s University of London, London SW17 0RE, UK; 3Department of Regional Health Research, University of Southern Denmark, 5230 Odense, Denmark; 4IMER Registry, Centre for Clinical and Epidemiological Research, University of Ferrara and Azienda Ospedaliero Universitario di Ferrara, 44121 Ferrara, Italy; 5Rare Diseases Research Unit, Foundation for the Promotion of Health and Biomedical Research in the Valencian Region, 46020 Valencia, Spain; 6Unit of Epidemiology of Rare Diseases and Congenital Anomalies, Institute of Clinical Physiology, National Research Council, 56124 Pisa, Italy; 7THL Finnish Institute for Health and Welfare, Department of Knowledge Brokers, 00271 Helsinki, Finland; 8Karolinska Institutet, Department of Molecular Medicine and Surgery, 171 77 Stockholm, Sweden; 9Faculty of Life and Health Sciences, Ulster University, Belfast BT15 1AP, UK; 10Faculty of Medicine, Health & Life Sciences, Swansea University, Swansea SA2 8PP, UK; 11Malformation Monitoring Centre Saxony-Anhalt, Medical Faculty, Otto-von-Guericke-University Magdeburg, 39106 Magdeburg, Germany; 12Congenital Anomaly Register & Information Service for Wales (CARIS) Public Health Knowledge and Research, Public Health Wales, Swansea SA6 8DP, UK; 13Section of Epidemiology, Department of Public Health, University of Copenhagen, 1353 Copenhagen, Denmark

**Keywords:** rare congenital anomalies, morbidity, length of stay, surgery

## Abstract

Little is known about morbidity for children with rare structural congenital anomalies. This European, population-based data-linkage cohort study analysed data on hospitalisations and surgical procedures for 5948 children born 1995–2014 with 18 rare structural congenital anomalies from nine EUROCAT registries in five countries. In the first year of life, the median length of stay (LOS) ranged from 3.5 days (anotia) to 53.8 days (atresia of bile ducts). Generally, children with gastrointestinal anomalies, bladder anomalies and Prune-Belly had the longest LOS. At ages 1–4, the median LOS per year was ≤3 days for most anomalies. The proportion of children having surgery before age 5 years ranged from 40% to 100%. The median number of surgical procedures for those under 5 years was two or more for 14 of the 18 anomalies and the highest for children with Prune-Belly at 7.4 (95% CI 2.5–12.3). The median age at first surgery for children with atresia of bile ducts was 8.4 weeks (95% CI 7.6–9.2) which is older than international recommendations. Results from the subset of registries with data up to 10 years of age showed that the need for hospitalisations and surgery continued. The burden of disease in early childhood is high for children with rare structural congenital anomalies.

## 1. Introduction

Around 2% of all liveborn children in Europe are diagnosed with a major congenital anomaly [1]. Mortality and morbidity for these children are higher than for children without congenital anomalies [2,3]. A recent study on specific congenital anomalies showed that children with severe congenital heart defects (CHD) and gastro-intestinal anomalies such as oesophageal atresia and intestinal atresia had a median length of stay in hospital of 20 days or more in the first year after birth [4]. Information on childhood morbidity for rare structural anomalies is more limited as there are many challenges in the research on rare anomalies such as difficulties in developing evidence-based treatment guidelines [5]. A recent study on survival up to 10 years of age for children with rare congenital anomalies showed varying survival rates with a low survival rate for children with holoprosencephaly (36%) and a high survival rate for children with Hirschsprung’s anomaly, accessory kidney or epispadias (>95%) [6].

EUROCAT is a network of population-based congenital anomaly surveillance registries in different regions of Europe [7,8]. The EUROlinkCAT study linked congenital anomaly registry data on liveborn children with all types of major congenital anomalies to healthcare databases to obtain information on mortality and morbidity [9]. The aim of this current study is to report overall morbidity for children with rare structural congenital anomalies measured as the risk of hospitalisation, the median number of days spent in hospital, the proportion of children with extended stays of 10 days or more, the proportion of children having surgery, the median number of surgical procedures and age at first surgery.

## 2. Material and Methods

This study is a European, population-based data-linkage cohort study, including data from nine EUROCAT registries (national and regional) in five countries (Appendix A). Details about the EUROCAT registries, hospital databases and linkage methods have been published elsewhere [4,9]. All live-born children with rare noncardiac structural congenital anomalies (livebirth prevalence <1 per 10,000 births) registered in the EUROCAT registries and born between 1995 (or the first year of the EUROCAT registry if later) and 2014 were included (termed “rare EUROCAT children”). Data on all live-born children without congenital anomalies born during the same time period and from the same population area covered by the registry were included as a reference population for comparison (termed “reference children”).

Data on hospitalisations and surgical procedures for all children up to their 10th birthday or the end of 2015 (i.e., at least one year of follow-up for all children), whichever came earlier, were obtained by electronic linkage to hospital databases. Hospital stays related to the birth were excluded from the analysis as in most countries almost all children are born in hospitals. The codes used for the exclusion of a hospital stay were ICD-10 codes Z37–Z39 (codes for the outcome of childbirth) or ICD-9 cm (clinical modification) codes V30–V39 (codes for the type of birth). All admissions on the day of birth or on day 1 that included additional diagnoses were included in the study.

Surgical procedures were coded according to the coding systems used in the national health record systems. Italy and Spain used ICD-9 cm for the study period, the United Kingdom (Wales and England) used OPCS-4 and Finland and Denmark used national adaptions of NCSP (NOMESCO Classification of Surgical Procedures). Three paediatricians independently reviewed all the codes from the extensive lists of surgery and procedure codes extracted from the three different coding systems and reached a consensus on which codes to define as surgery [9].

The results for each type of anomaly were pooled across registries and analysed for all children diagnosed with the anomaly. For the rare anomaly studies in EUROlinkCAT, it was not possible to analyse data on isolated anomalies only due to governance restrictions on the use of data based on small numbers in several of the databases providing data for the study.

### Statistical Analysis

A common data model was developed to standardise the linked variables from the local hospital databases in each registry [4,6,9]. This enabled all registries to run centrally written syntax scripts in Stata version 13 for linkage quality checks and morbidity analyses. No individual case data were shared as all analyses were performed locally using these linked datasets. The aggregate tables and analytic results produced in a common format were then sent to a Central Results Repository at Ulster University for collation and redistribution to the study team. Analyses were performed on age groups (<1 year, 1–4 years, 0–4 years and 0–9 years) separately. For all nine registries, hospitalisations and surgery were examined in three age groups: <1 year, 1–4 years and 0–4 years. For six of the nine registries, data on hospitalisations and surgery between 5 and 9 years of age were also available (only children born in 1995–2005 reached the age of 10 years before the end of 2015) and for these registries, the outcomes were also examined at 0–9 years.

Kaplan–Meier survival analysis was used to estimate the proportions of children experiencing each outcome of interest (admitted to hospital, staying 10 days or more, undergoing surgery), which accounted for censoring due to early exit from the study. The median and interquartile range of the number of days spent in hospital, the number of surgical procedures and age at first surgery by anomaly subgroup were reported by each registry. Details for the meta-analytic methods to combine proportions and medians across registries can be found in a previous publication [4].

## 3. Results

The study included data on 5948 children with 18 rare congenital anomalies from nine EUROCAT registries in five western European countries (Finland, Denmark, the United Kingdom, Italy and Spain) (Table 1). The number of children born with each anomaly ranged from 27 children with Prune-Belly to 867 children with unilateral renal agenesis, of whom, 21 with Prune-Belly and 816 with unilateral renal agenesis reached at least one year of age during the follow-up period. The proportion of children with rare congenital anomalies admitted to hospital in the first year after birth was very high at 85% or more. The proportion of term-born children with a hospital stay of 10 days or more differed by anomaly type but was very high for children with atresia of the bile ducts (94%), bladder exstrophy (87%) and annular pancreas (76%). The proportion of children admitted to hospital whilst aged 1–4 was at or above 47% for all types of anomalies, but the proportion of children with long stays (10 days or more) was much lower compared to the first year of life. A high proportion of children with arhinencephaly/holoprosencephaly had hospital stays of 10 days or more.

The median length of stay in the first year ranged from 3.5 days for children with anotia to 53.8 days for children with atresia of the bile ducts (Table 1). Generally, children with gastrointestinal anomalies, bladder anomalies and Prune-Belly had the longest stay in hospital during the first year. At ages 1–4, the median length of stay per year was under 3 days for the majority of anomalies.

The proportions of children having any surgery are presented in Table 2. The proportion of children having surgery before age 5 ranged from 40–100% and was lowest in children with anotia, unilateral renal agenesis and situs inversus. For 10 of the 18 rare congenital anomalies, the proportion with surgery was higher in the first year compared to ages 1–4, and for eight anomalies, the proportion with surgery was higher at ages 1–4 compared to the first year of life (e. g. agenesis of corpus callosum, anotia, unilateral renal agenesis). Only 53% of children with encephalocele had surgery within the first year and 41% at age 1–4 years.

The median number of surgical procedures before age 5 years was 2.0 or higher for 14 of the 18 anomalies. The lowest median number of surgical procedures was for children with encephalocele at 1.3. Children with Prune-Belly and bladder exstrophy had the highest median number of surgical procedures. The median age at first surgery was less than one week for children with anomalies of intestinal fixation, annular pancreas and bladder exstrophy. For children with atresia of the bile ducts, the median age at surgery was 8.4 weeks (95% CI 7.6–9.2 weeks). The highest median age at first surgery was for children with anotia (41.8 weeks, 95% CI 0.0–85.1), but the confidence interval was very wide due to the relatively small number of cases.

Six registries were able to provide data on the median length of stay and surgery for the age group 0–9 years, and 5013 children were included (Table 3). The overall pattern is the same as for children aged less than 5 years, but the median number of surgical procedures was higher, indicating that the burden of disease for these children continues after age 5. The median length of stay per year for these 10 years was very high for children with Prune-Belly, atresia of the bile ducts and arhinencephaly/holoprosencephaly (9.6, 8.0 and 3.9 days, respectively). Children with congenital arthrogryposis multiplex congenita had a median length of stay of 2.4 days per year, 89% had surgery performed and the median number of surgical procedures was 4.4 before age 10 years. The median number of surgical procedures for children with anophthalmos/microphthalmos and congenital glaucoma was 4.2 and 4.4 for the first 10 years.

## 4. Discussion

This study showed that most children with these rare structural congenital anomalies have long stays in hospital within the first year after birth and most children have more than one surgical procedure performed before age 5 years. We do not know if all surgical procedures performed were directly related to the anomaly. However, to describe the full burden of disease for these children it is necessary to include all surgical procedures, including those that are a consequence of the anomaly or complications to the surgical correction of the anomaly as well as hospital stays and surgical procedures that are unrelated to the anomaly.

Children with structural anomalies may have associated anomalies in other organ systems or an associated genetic abnormality. Overall, 76% of all congenital anomaly cases in the EUROCAT central database have an isolated structural anomaly [10], but this proportion is expected to be a little higher for liveborn children. The EUROlinkCAT study on the length of stay in hospital for children with more common congenital anomalies showed that children with specific isolated anomalies had a median length of stay in the first year that was approximately 1 day shorter than for all children with the anomaly [4]. Regarding the rare anomalies included in this study, 21% of children with Hirschsprung’s disease have associated anomalies [11] and 13.6% of children with congenital glaucoma have co-morbidities [12]. In general, we would expect morbidity to be less severe and show less variation for children with isolated anomalies only.

For the four gastrointestinal anomalies in this study, surgery is needed for survival, but surgery was not reported for all the children. Some children may die before surgery, although the first week survival for these four anomalies was rather high in the EUROlinkCAT study at 95.8–99.8% [6]. There may be underreporting of surgery in the hospital databases and there may be problems with the registration of surgery in the first days after birth due to delays in having a personal ID number or name. For encephalocele, only half of the children had surgery within the first year. This may be because encephalocele is associated with severe chromosomal defects, Meckel-Gruber syndrome or other severe cerebral anomalies, and therefore, palliative treatment without surgery could be the optimum management.

Children with arhinencephaly/holoprosencephaly had long hospital stays continuing up to age 10 years although only three out of four children had any surgery performed within these 10 years. The extended hospital stays may be due to developmental problems associated with this anomaly, including epilepsy [13]. The EUROlinkCAT survival study found a 10-year survival of children with arhinencephaly/holoprosencephaly of only 36% [6]. These two EUROlinkCAT studies document very high morbidity and mortality for children born with arhinencephaly/holoprosencephaly, although some children with a less severe form of the anomaly may have a normal life [13].

Children with the eye anomalies congenital glaucoma and anophthalmos/microphthalmos had rather short median lengths of stay compared with the other anomaly groups included in this study, but the median numbers of surgical procedures within the first 10 years were high at 4.2 and 4.4, respectively. A Danish study including 40 birth cohorts found a median age at first surgery for children with congenital glaucoma at 21 weeks and with some diagnostic delays from first suspicion to the final diagnosis by an ophthalmologist [12]. In this European study, the median age at first surgery was 10 weeks. We do not know if the distribution of unilateral and bilateral glaucoma was the same in these two populations.

In line with the previous study on the length of stay for more common congenital anomalies, this study indicated a long median length of stay for children with gastrointestinal anomalies. For children with Hirschsprung’s disease, we report a median length of stay of 31.2 days in the first year and 1.7 days per year at age 1–4 years which is lower than the total median length of stay of 49.7 days in children up to 5 years found in a similar study from Australia [3]. Additionally, a study from the UK documented high morbidity in the first years [14]. There is no international consensus on the best surgical treatment for children with Hirschsprung’s disease [15].

The median age at first surgery for children with atresia of the bile ducts was 8.4 weeks. This means that half of the children in this study had their first surgery after 8 weeks. The general recommendation is that surgery is performed before age 8 weeks for better long-term outcomes without liver transplantation [16,17]. The median length of stay of 8.0 days per year up to 10 years found in our study indicates continuous high morbidity for these children after the initial surgery.

Our results for arthrogryposis multiplex congenita were based on 169 children. An American database studying the outcome of children with this diagnosis includes data on 40 children with the aim of extending the database to 400 children [18]. Treatment is early and intensive physiotherapy combined with surgery. In our data, the median age at first surgery was 25.4 weeks, and the median number of surgical procedures was 3.3 within the first 5 years and 4.4 before age 10, indicating increased morbidity throughout childhood. The EUROlinkCAT study on survival found a 10-year survival at 69.4% for children with arthrogryposis multiplex congenita, confirming the severity of this anomaly [6].

Although our study population covers 2 million births there was a limited number of children with some of the anomalies in each registry. We tried to overcome these problems in the way we defined the study questions and inclusions. We believe we have demonstrated a method to obtain pooled and non-identifying information on morbidity for these patients by linking data from patient registries for rare diseases to hospital databases across health systems and coding systems.

### Strengths and Limitations

The main strength of this study is the population-based setting covering all children and not, like many existing studies, only those referred to tertiary hospitals for treatment. The study was based on EUROCAT congenital anomaly registries with their high level of case ascertainment and use of standardised definitions and coding of congenital anomalies across the registries. An important limitation is the use of surgery codes from different coding systems that may not be completely comparable. There may be some underreporting of surgery in the hospital databases, although that is unlikely to be significant given that a key function of these databases is for the financial reimbursement of costly treatments such as surgery. It is also a limitation that we were not able to examine children with an isolated anomaly only. This was due to the small numbers in several registries and restrictions on the release of results based on only a few children. However, as the data are from a population-based cohort, the presence of associated anomalies reflects the expected occurrence of anomalies in future births, and, therefore, the results can be considered an unbiased estimate of morbidity for children with these rare anomalies.

## 5. Conclusions

Children with rare structural congenital anomalies have a high burden of disease within the first 10 years of life with many children having long hospital stays and several surgical procedures performed. It is important to support the child and the family to cope with the diagnosis so that the child can have as normal a childhood as possible. The study covered a birth population of 2 million births, but even so, there were low numbers in some analyses, reflecting the issues in research on rare conditions. Wider international collaboration is needed to combine results from databases across the globe to inform healthcare professionals and families of the immediate outlook and long-term prognoses for children with rare congenital anomalies.

## Figures and Tables

**Table 1 ijerph-20-04387-t001:** Proportion of children admitted to hospital, proportion of term-born children with hospital stays of 10 days or more and median length of stay for children with 18 rare congenital anomalies and for reference children by age group.

	AGE < 1				AGE 1–4 Years			
	No of Children Born	% Admitted ^a^ (95% CI)	% Admitted > 10 Days ^b^ (95% CI)	Median length of Stay ^c^ (95% CI)	No of Children Reaching 1 Year of Age	% Admitted ^a^ (95% CI)	% Admitted > 10 Days ^b^ (95% CI)	Median Length of Stay per Year ^c^ (95% CI)
Reference Children	1,960,272	30% (25–36)	1% (1–2)	3.0 (2.1–3.9)	1,935,199	23% (17–31)	1% (1–1)	0.4 (0.3–0.6)
Encephalocele	136	93% (78–98)	31% (22–41)	9.7 (7.2–12.1)	110	65% (44–79)	9% (4–18)	1.2 (0.6–1.8)
Arhinencephaly/holoprosencephaly	94	85% (57–96)	58% (39–74)	20.6 (11.8–29.5)	50	99% (88–100)	36% (18–55)	4.4 (1.7–7.1)
Anomalies of corpus callosum	686	94% (88–97)	43% (36–51)	13.6 (10.0–17.1)	590	73% (64–81)	17% (13–20)	2.3 (1.3–3.2)
Anophthalmos ^d^/microphthalmos	355	92% (84–97)	30% (21–41)	9.7 (6.0–13.5)	292	78% (69–85)	10% (6–16)	1.2 (0.7–1.7)
Congenital glaucoma	167	95% (86–98)	14% (9–21)	8.1 (6.8–9.4)	163	94% (81–98)	8% (4–13)	0.9 (0.4–1.4)
Anotia	68	97% (81–100)	14% (5–28)	3.5 (1.6–5.4)	64	63% (47–76)	1% (0–10)	0.5 (0.3–0.8)
Cystic adenomatous malformation of lung	370	91% (83–95)	18% (7–34)	8.6 (2.8–14.5)	345	48% (38–56)	1% (0–4)	0.5 (0.2–0.7)
Hirschsprung’s disease	687	97% (93–99)	74% (66–81)	31.2 (25.4–36.9)	666	74% (69–79)	15% (12–19)	1.7 (1.2–2.2)
Anomalies of intestinal fixation	502	98% (96–99)	67% (58–74)	19.9 (15.6–24.1)	430	52% (38–65)	11% (7–15)	0.7 (0.4–1.0)
Atresia of bile ducts	171	99% (97–100)	94% (86–97)	53.8 (46.8–60.8)	148	84% (71–91)	39% (29–49)	6.2 (4.3–8.1)
Annular pancreas	99	99% (94–100)	76% (47–90)	22.2 (17.3–27.0)	94	62% (20–86)	5% (1–15)	0.5 (0.2–0.9)
Unilateral renal agenesis	867	85% (76–90)	18% (13–23)	5.4 (3.3–7.4)	816	55% (49–61)	5% (3–7)	0.7 (0.5–0.9)
Accessory kidney	810	90% (83–94)	18% (11–26)	6.4 (4.7–8.1)	779	62% (51–72)	4% (2–7)	0.7 (0.5–0.9)
Bladder exstrophy	130	99% (96–100)	87% (77–93)	27.0 (20.4–33.6)	119	92% (82–96)	19% (4–43)	3.7 (2.0–5.3)
Posterior urethral valves	414	99% (97–100)	51% (45–57)	20.3 (14.2–26.4)	378	84% (70–92)	7% (4–11)	1.0 (0.7–1.3)
Prune Belly	27	98% (80–100)	63% (22–87)	44.3 (3.8–84.9)	21	99% (90–100)	29% (4–63)	13.4 (5.2–21.7)
Situs inversus	196	92% (86–95)	45% (33–55)	12.5 (8.2–16.7)	173	65% (50–77)	13% (5–23)	0.6 (0.4–0.9)
Arthrogryposis multiplex congenita	169	95% (89–98)	44% (33–55)	14.9 (9.9–19.8)	131	88% (77–94)	13% (4–28)	2.0 (1.1–2.8)

^a^: Number of children ever hospitalised in age period. ^b^: proportion of children born at term with at least one hospital stay of 10 days or more. ^c^: Median length of stay in days per year. Calculated only among children hospitalised in age period. Registries with <3 cases in subgroup not included. ^d^: 76 with a diagnosis of anophthalmos.

**Table 2 ijerph-20-04387-t002:** The percentage of children with rare congenital anomalies having any surgery by age group, median number of surgical procedures before age 5 years and median age at first surgery.

Type of Anomaly	Number of Children Born	Percentage with Any Surgery (95% CI)			Median Number of Surgical Procedures in the First 5 Years (95% CI)	Median Age in Weeks at First Surgery in the First 5 Years (95% CI)
		<1 year	1–4 years	<5 years		
Reference children	1,960,272	0.8% (0.5–1.1)	6.2% (4.0–9.2)	7.0% (4.6–10.0)	1.0 (1.0–1.0)	151.0 (130.1–171.9)
Encephalocele	136	53% (35–68)	41% (26–55)	78% (58–90)	1.3 (0.9–1.8)	3.5 (0.0–7.2)
Arhinencephaly/holoprosencephaly	94	36% (14–58)	55% (30–74)	63% (32–83)	3.6 (2.2–5.0)	13.3 (7.3–19.4)
Anomalies of corpus callosum	686	31% (24–38)	41% (33–50)	51% (41–61)	2.3 (1.8–2.7)	21.8 (11.6–31.9)
Anophthalmos/microphthalmos	355	45% (39–51)	49% (37–60)	62% (55–68)	3.4 (2.4–4.5)	17.0 (13.0–20.9)
Congenital glaucoma	167	70% (58–79)	55% (39–68)	81% (68–89)	3.3 (2.2–4.4)	10.0 (5.9–14.1)
Anotia	68	23% (12–36)	31% (9–57)	46% (28–62)	2.0 (0.7–3.3)	41.8 (0.0–85.1)
Cystic adenomatous malformation of lung	370	50% (29–68)	24% (14–35)	71% (53–83)	1.4 (1.0–1.8)	32.8 (21.2–44.4)
Hirschsprung’s disease	687	85% (77–91)	45% (38–52)	94% (88–97)	3.5 (3.1–3.9)	3.7 (1.3–6.1)
Anomalies of intestinal fixation	502	85% (81–89)	21% (14–29)	90% (87–93)	2.6 (2.1–3.0)	0.8 (0.5–1.0)
Atresia of bile ducts	171	89% (83–93)	54% (44–63)	92% (84–96)	3.6 (2.6–4.7)	8.4 (7.6–9.2)
Annular pancreas	99	92% (68–98)	17% (2–46)	93% (76–98)	1.9 (1.3–2.5)	0.4 (0.1–0.6)
Unilateral renal agenesis	867	23% (17–29)	28% (23–34)	40% (34–46)	2.2 (1.6–2.7)	23.9 (11.5–36.4)
Accessory kidney	810	36% (28–44)	38% (33–43)	57% (51–63)	1.8 (1.5–2.2)	27.3 (15.1–39.5)
Bladder exstrophy	130	86% (72–93)	82% (69–90)	96% (82–99)	5.6 (3.7–7.5)	0.3 (0.2–0.4)
Posterior urethral valves	414	75% (67–82)	46% (36–55)	86% (81–89)	2.4 (1.9–2.8)	2.9 (1.7–4.1)
Prune Belly	27	67% (36–85)	99% (90–100)	100% (94–100)	7.4 (2.5–12.3)	20.5 (0.0–61.3)
Situs inversus	196	33% (20–46)	27% (16–40)	41% (27–54)	3.2 (2.5–4.0)	1.9 (0.0–4.0)
Arthrogryposis multiplex congenita	169	61% (52–69)	67% (53–78)	83% (73–89)	3.3 (2.7–4.0)	25.4 (18.3–32.6)

**Table 3 ijerph-20-04387-t003:** Median length of stay and surgical procedures of children up to the age of 10 years for selected registries with follow-up of up to 10 years.

Type of Anomaly	Number of Children Born	Number Hospitalised	Number Registries	Median Length of Stay in Days per Year (95% CI)	Percentage with Any Surgery in the First 10 Years (95% CI)	Median Number of Surgical Procedures in the First 10 Years (95% CI)
Encephalocele	118	99	5	1.2 (0.7–1.8)	82% (71–90)	1.8 (1.2–2.3)
Arhinencephaly/holoprosencephaly	86	62	5	3.9 (2.6–5.3)	76% (25–95)	4.0 (2.1–6.0)
Anomalies of corpus callosum	569	545	6	1.6 (0.7–2.6)	66% (56–74)	3.1 (2.7–3.6)
Anophthalmos/microphthalmos	293	271	6	0.8 (0.4–1.2)	71% (64–77)	4.2 (3.6–4.8)
Congenital glaucoma	120	117	5	1.4 (1.1–1.8)	82% (67–91)	4.4 (3.4–5.5)
Anotia	55	44	4	0.4 (0.2–0.5)	67% (48–80)	2.8 (0.9–4.7)
Cystic adenomatous malformation of lung	323	299	5	0.6 (0.5–0.8)	64% (44–78)	1.3 (0.8–1.7)
Hirschsprung’s disease	587	582	6	3.5 (2.9–4.1)	96% (92–98)	4.0 (3.3–4.7)
Anomalies of intestinal fixation	440	420	6	2.0 (1.4–2.6)	92% (87–95)	2.3 (1.9–2.7)
Atresia of bile ducts	134	131	4	8.0 (6.0–10.1)	99% (88–100)	3.2 (1.9–4.5)
Annular pancreas	81	77	4	2.4 (1.7–3.0)	88% (78–94)	2.0 (1.6–2.4)
Unilateral renal agenesis	653	557	6	0.6 (0.4–0.8)	47% (38–56)	2.6 (1.9–3.2)
Accessory kidney	744	690	6	0.9 (0.8–1.0)	66% (62–70)	1.8 (1.4–2.2)
Bladder exstrophy	114	112	6	5.2 (3.5–7.0)	99% (84–100)	6.8 (4.3–9.2)
Posterior urethral valves	365	353	6	2.0 (1.7–2.3)	88% (83–91)	2.5 (1.9–3.2)
Prune Belly	25	16	2	9.6 (1.9–17.3)	100% (98–100)	7.8 (3.9–11.7)
Situs inversus	159	151	6	1.5 (0.9–2.1)	55% (42–66)	4.0 (2.5–5.5)
Arthrogryposis multiplex congenita	147	136	5	2.4 (1.6–3.1)	89% (78–95)	4.4 (3.4–5.4)

## Data Availability

The data that support the findings of this study are available, but restrictions apply to the availability of these data, which were used under license for the current study and so are not publicly available. Data are, however, available from the authors after the permission of the participating registries of congenital anomalies.

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
