# Peer review of "Hospital Length of Stay and Surgery among European Children with Rare Structural Congenital Anomalies—A Population-Based Data Linkage Study"

_ijerph, 2023, doi:10.3390/ijerph20054387_

Round 1

Reviewer 1 Report

I read with interest the manuscript entitled "Hospital length of stay and surgery among European children with rare structural congenital anomalies – A population-based data linkage study"

The rules for the surgery definition (Appendix Table 2) are questionable. Who defined the table and based on which parameters? For which anomaly do you have "Removal of foreign bodies from the bronchus, lungs, and esophagus" Please clarify.

The particular problem of this manuscript is that it observes the number of operations, length of hospitalization, etc. in the context of one anomaly, and we know that many of them are associated.

You mention a lot of information in the text of the manuscript, which is contained in the tables. What is stated in the tables should not be mentioned again in the manuscript!

When drawing a conclusion from a study like this, you need to know whether and which surgical procedures were directly related to the anomaly. Consequently, the results are very questionable!

Ultimately, the data conceived in this way are part of the European Union statistical data that are publicly available. What is the ultimate purpose of the manuscript? What new did you show? It is common knowledge that children with congenital anomalies are significantly burdened than the population without anomalies.

If you had compared the data between the states included in the register, if you had looked at the financial burdens between the states for each individual anomaly, etc., you could draw conclusions and plan interventions.

The manuscript conceived in your way seems as if we are only reading the data included in the register, without conclusions that would encourage the reader to think.

Author Response

I read with interest the manuscript entitled "Hospital length of stay and surgery among European children with rare structural congenital anomalies – A population-based data linkage study"

The rules for the surgery definition (Appendix Table 2) are questionable. Who defined the table and based on which parameters? For which anomaly do you have "Removal of foreign bodies from the bronchus, lungs, and esophagus" Please clarify.

The rules for the surgery definition were based on frequency lists from each registry for all procedures performed in all children less than 10 years of age during the study period. We expect that the surgical procedure mentioned above has mainly been performed in children without congenital anomalies (98% of the children in the population). As this paper includes very limited results for the reference children, we have deleted Appendix Table 2 to avoid confusion. The table will be included in a more general surgical paper form EUROlinkCAT.

To the editorial office: please delete Appendix Table 2 in the submitted file.

The particular problem of this manuscript is that it observes the number of operations, length of hospitalization, etc. in the context of one anomaly, and we know that many of them are associated.

We agree with the reviewer that some of these children have more than one anomaly and that these children may have more surgical procedures and longer hospital stays. We know from the EUROCAT database that this applies to 24% for all fetuses for most types of anomalies with the highest proportion for terminations of pregnancy. As this study included liveborn children and very rare anomalies only, we had severe limitations in the release of the data from the hospital databases. We therefore decided to ask for results for all children with the anomalies for the study, as the slightly higher numbers would have less restrictions for release of results. In our study on more common anomalies, we found that the children with associated anomalies had a median length of stay in the first year that was approx. 1 day longer than for the children with isolated anomalies (Urhoj et al 2022). We have discussed these issues in the beginning of the Discussion section.

You mention a lot of information in the text of the manuscript, which is contained in the tables. What is stated in the tables should not be mentioned again in the manuscript!

Thank you. We have deleted some numbers in the Result section. Our aim when writing the Result section was to mention each anomaly included in the study. We also think that the interval/range of results should be presented in the text.

When drawing a conclusion from a study like this, you need to know whether and which surgical procedures were directly related to the anomaly. Consequently, the results are very questionable!

We agree with the reviewer that not all surgical procedures are for correction of the anomaly. Children with congenital anomalies may have surgical procedures performed that are a consequence of the anomaly such as surgery for gastrostomy, surgery for wound infections etc. They will also undergo the same surgical procedures as children without anomalies such as surgery for appendicitis, adenotomy and others. Our aim was to give results for the overall burden of disease for the child and not for the specific anomaly/organ system and we have added information for children without congenital anomalies for comparison.

Ultimately, the data conceived in this way are part of the European Union statistical data that are publicly available. What is the ultimate purpose of the manuscript? What new did you show? It is common knowledge that children with congenital anomalies are significantly burdened than the population without anomalies.

This study is novel in that data from congenital anomaly registries were linked to routine hospital data. Our earlier work has shown that the reporting of CAs in such hospital data is not sufficiently accurate for research to be conducted in these rare anomalies using only the publicly available hospital data. We agree that it is common knowledge that morbidity is higher for children with congenital anomalies, but there is very little published data quantifying this excess morbidity for specific rare anomalies and we have now provided this information. There were too few children in each registry to publish their own results and therefore European collaboration was needed.

If you had compared the data between the states included in the register, if you had looked at the financial burdens between the states for each individual anomaly, etc., you could draw conclusions and plan interventions.

The congenital anomalies included in this paper are all very rare and, as stated in limitations, numbers were too small to allow publishing results by individual registry and region. See also our reply to the comment about associated anomalies above.

The manuscript conceived in your way seems as if we are only reading the data included in the register, without conclusions that would encourage the reader to think.

We think that we have improved our paper based on the reviewers’ comments.

Reviewer 2 Report

The paper is well written and enjoyable to read. The main question addressed by the research is morbidity for children with rare structural congenital anomalies.

The study describes from the age of the first procedure, as well as the number of procedures and length of stay in a large sample. In general, the studies are few, with smaller and local samples. The methodology is adequate to the objectives of the study.

In the tables, the lines between the items could be removed and the spacing between lines could be improved.

The conclusions are consistent with the evidence and arguments
presented and do they address the main question posed

The impact of structural congenital anomalies on morbidity is not clearly known. It is known that it has a negative impact, but this is not clear in terms of procedures and length of stay. The study focuses on 18 structural anomalies and involves registries from 5 countries, describing how the anomalies studied impact on the morbidity of those children.

The only aspect that was not clear to me was whether the number of children indicated in each age group was cumulative or not. I think it would be interesting to make this aspect very clear.

Author Response

The only aspect that was not clear to me was whether the number of children indicated in each age group was cumulative or not. I think it would be interesting to make this aspect very clear.

We apologise for the lack of clarity, and we have amended the text to make it clearer that number of children aged < 1 is the number of children who were born, and the number of children aged 1-4 is the number of children who reached at least one year of age during the follow-up period.

Reviewer 3 Report

I want to congratulate the author for this very comprehensive manuscript, original and very well written. I would suggest a larger reference list. 

Author Response

I want to congratulate the author for this very comprehensive manuscript, original and very well written. I would suggest a larger reference list. 

Thank you for the positive review of our paper. We have added 3 more references.

Reviewer 4 Report

The article offers information regarding medical management approaches of children with rare structural congenital anomalies from Europe. Research design is appropriate and the statistical analysis is adequate. Conclusions are supported by the results and discussion is to the point. 

However, there is the following point for corrections prior to publication:

Please check he correctness of the English language

Author Response

However, there is the following point for corrections prior to publication:

Please check he correctness of the English language

Thank you for the positive review of our paper. We have corrected some spelling errors and have also improved the English in several paragraphs.

Round 2

Reviewer 1 Report

Thank you for your answers.

Even after the submitted revision of the manuscript, I believe that the manuscript in this form does not exude the quality of an original scientific journal worthy of publication in journals with a high impact factor such as yours.As the authors do not have clear criteria for the surgery definition, the scientific validity becomes questionable. The explanation given is questionable to say the least! The authors themselves state in their response that there are serious limitations of the study. When conducting a study like this, it is clear that you must have data on specific surgical interventions exclusively related to children with anomalies compared to children without them. These are the basic postulates of the target and control groups! The presented numbers are not small, as the authors state. With adequate statistical methods, they can be compared without any problems, from which conclusions could be drawn and interventions planned.

Kind regards

Author Response

To the Editor: please also see attachment.

Reviewer

Comments and Suggestions for Authors

Even after the submitted revision of the manuscript, I believe that the manuscript in this form does not exude the quality of an original scientific journal worthy of publication in journals with a high impact factor such as yours. As the authors do not have clear criteria for the surgery definition, the scientific validity becomes questionable. The explanation given is questionable to say the least! The authors themselves state in their response that there are serious limitations of the study. When conducting a study like this, it is clear that you must have data on specific surgical interventions exclusively related to children with anomalies compared to children without them. These are the basic postulates of the target and control groups!

The main aim of our study was to report overall morbidity for children with rare anomalies and not to report the specific surgical procedures that were performed.

We understand that the reviewer has some concern about the criteria for the surgery definition. We believe we have clear and valid definitions of the surgical procedures and that they are sufficient for subsequent researchers to replicate our analysis.

All hospital databases included in the study had a variable for surgery, but looking at the variable for surgery in each database made it clear to us that we needed to clean the data before comparing and pooling the results. We worked extensively with defining what was surgery within each database. Each country extracted ALL codes for procedures performed on children during the study period for review. It was also clear that surgeons in different hospitals did not use the same codes for the same surgical procedures and some children had rather unspecified surgery codes. As a result, we worked extensively with defining what was surgery within each database.The full lists of codes defining surgical procedures could be made available on request in order for readers to evaluate these decisions in more detail.

We believe our analysis is valid , but we do have to accept the limitations of using health care data rather than data generated for research purposes.

The presented numbers are not small, as the authors state. With adequate statistical methods, they can be compared without any problems, from which conclusions could be drawn and interventions planned.

We agree that for some anomalies the total number of children is not small. However, the data come from nine geographical areas and therefore this additional heterogeneity must be allowed for in the analysis. We believe we have performed the correct meta-analysis and now we have edited the tables to provide the 95% confidence intervals on all the estimates, which will allow the readers to compare the anomalies as they wish and draw their own conclusions.

Round 3

Reviewer 1 Report

Thank you for the answers, but please do not formulate conclusions in general. There is no doubt that children with severe congenital anomalies will have to stay in the hospital for a longer period of time and undergo several surgical procedures. I expect you, based on the data from the database, to give concrete conclusions and possible interventions for this population of children. Compare data between countries and see where decisive interventions need to be implemented to improve healthcare delivery. Who works better and who works worse? Readers should not be left to draw their own conclusions. Also, do not cite references in the conclusion, as this is not customary in high-quality scientific articles. The manuscript still gives the impression of general citations of data from the database without specific and high-quality conclusions that will lead to the improvement of health care provision for this target population of children.

Author Response

Thank you for the answers, but please do not formulate conclusions in general. There is no doubt that children with severe congenital anomalies will have to stay in the hospital for a longer period of time and undergo several surgical procedures. I expect you, based on the data from the database, to give concrete conclusions and possible interventions for this population of children.

We can see that the reviewer would like to see concrete conclusions and possible interventions for these children. However, it is beyond the scope of the data and hence any such comments would be purely speculative, and we do not feel these speculations should be added to the paper. 

Compare data between countries and see where decisive interventions need to be implemented to improve healthcare delivery. Who works better and who works worse? Readers should not be left to draw their own conclusions.

Again, unfortunately due to the small numbers of cases within each registry the data is not sufficient to enable us to make statistically significant comparisons between countries. Such comparisons would also require adjusting for potential confounders, which again is not possible given the data collected.

 Also, do not cite references in the conclusion, as this is not customary in high-quality scientific articles.

We have removed the references in the conclusion.

 The manuscript still gives the impression of general citations of data from the database without specific and high-quality conclusions that will lead to the improvement of health care provision for this target population of children.

We do not agree with this statement – the other reviewers found the study was informative.